# Resistance to MET/VEGFR2 Inhibition by Cabozantinib Is Mediated by YAP/TBX5-Dependent Induction of FGFR1 in Castration-Resistant Prostate Cancer

**DOI:** 10.3390/cancers12010244

**Published:** 2020-01-19

**Authors:** Filippos Koinis, Paul Corn, Nila Parikh, Jian Song, Ioulia Vardaki, Ioanna Mourkioti, Sue-Hwa Lin, Christopher Logothetis, Theocharis Panaretakis, Gary Gallick

**Affiliations:** 1Department of Genitourinary Medical Oncology, The University of Texas MD Anderson Cancer Center, Houston, TX 77030, USA; phillipkoinis@gmail.com (F.K.); PCorn@mdanderson.org (P.C.); nparikh@mdanderson.org (N.P.); jhsong@mdanderson.org (J.S.); IVardaki1@mdanderson.org (I.V.); IMourkioti@mdanderson.org (I.M.); slin@mdanderson.org (S.-H.L.); clogothe@mdanderson.org (C.L.); 2Translational Molecular Pathology, The University of Texas MD Anderson Cancer Center, Houston, TX 77030, USA

**Keywords:** prostate cancer, resistance, c-MET, FGFR1, YAP, TBX5, cabozantinib

## Abstract

The overall goal of this study was to elucidate the role of FGFR1 induction in acquired resistance to MET and VEGFR2 inhibition by cabozantinib in prostate cancer (PCa) and leverage this understanding to improve therapy outcomes. The response to cabozantinib was examined in mice bearing patient-derived xenografts in which FGFR1 was overexpressed. Using a variety of cell models that reflect different PCa disease states, the mechanism underpinning FGFR1 signaling activation by cabozantinib was investigated. We performed parallel investigations in specimens from cabozantinib-treated patients to confirm our in vitro and in vivo data. FGFR1 overexpression was sufficient to confer resistance to cabozantinib. Our results demonstrate transcriptional activation of FGF/FGFR1 expression in cabozantinib-resistant models. Further analysis of molecular pathways identified a YAP/TBX5-driven mechanism of FGFR1 and FGF overexpression induced by MET inhibition. Importantly, knockdown of YAP and TBX5 led to decreased FGFR1 protein expression and decreased mRNA levels of FGFR1, FGF1, and FGF2. This association was confirmed in a cohort of hormone-naïve patients with PCa receiving androgen deprivation therapy and cabozantinib, further validating our findings. These findings reveal that the molecular basis of resistance to MET inhibition in PCa is FGFR1 activation through a YAP/TBX5-dependent mechanism. YAP and its downstream target TBX5 represent a crucial mediator in acquired resistance to MET inhibitors. Thus, our studies provide insight into the mechanism of acquired resistance and will guide future development of clinical trials with MET inhibitors.

## 1. Introduction

Prostate cancer (PCa) is a leading cause of cancer morbidity and mortality in men worldwide, with an estimated 174,650 new cases and 31,620 deaths in 2019 in the United States [1]. Treatment has been proven effective for the early stage of this hormone-sensitive disease, but PCa eventually progresses to an androgen-independent state, leading to bone and soft tissue metastases. Recently, several agents have been approved that prolong patients’ overall survival (OS) [2]. Nevertheless, metastatic castration-resistant PCa (mCRPC) remains incurable. As many receptor tyrosine kinases (RTKs) are overexpressed in mCRPC and predictive of poor prognosis [3], inhibitors of RTKs (Tyrosine kinase inhibitors, TKIs), such as c-MET (mesenchymal to epithelial transition, proto-oncogene), VEGFR2 (vascular endothelial growth factor receptor 2), FGFRs (Fibroblast growth factor receptors), PDGFR (platelet-derived growth factor receptors), HER2/neu (human epidermal growth factor receptor 2/neu), and IGF-1R (insulin-like growth factor 1 receptor) have undergone many clinical trials in PCa [3]. Nonetheless, despite a valid preclinical rationale, none have succeeded in achieving FDA approval for the treatment of patients with CRPC. A recent example of the difficulty in translating murine investigations into clinically meaningful responses is cabozantinib, an oral multikinase inhibitor with potent activity against phospho-MET and phospho-VEGFR2, which demonstrated clinical and radiological responses in patients with CRPC with bone metastasis [4]. Despite efficacy in individual patients, a phase III trial in unselected patients failed to demonstrate a survival benefit [5]. These findings can be explained by the fact that several tyrosine kinases can drive tumor progression in patients with CRPC. Elucidating the mechanisms of resistance to TKIs and leveraging that understanding to develop strategies to overcome the resistance will be of clinical significance.

Multiple mechanisms of resistance to molecularly targeted treatment have been described [6,7]. Acquired resistance to TKIs is often linked to overexpression of non-targeted RTKs that bypass the point of inhibition and activate similar downstream pathways. We recently showed that FGFR1 was increased upon long-term treatment of patient-derived xenografts (PDXs) with cabozantinib, and knockdown of MET in cell lines increased FGFR1 expression [8]. Previous studies in PCa have shown that aberrant activation of the FGF/FGFR1 axis is a driving force in carcinogenesis. While FGFR1 expression in the normal prostate gland [9] is limited to the stroma, aberrant expression of FGFR1 in epithelial cells is a common trait for PCa. Acevedo et al. have shown that activation of FGFR1 in a prostate mouse model (JOCK1) was associated with progression to adenocarcinoma and epithelial-to-mesenchymal transition [9]. Preclinical data have demonstrated that the FGF/FGFR1 axis is a key mediator of angiogenesis [10,11] and PCa dissemination [10,11], whereas targeting this pathway inhibited both tumor growth and blood vessel formation [12]. FGFR1 overexpression in patient samples is associated with shorter time to the development of castration resistance [13] and is indicative of a more aggressive clinical phenotype [13,14]. However, the mechanisms that lead to FGFR1 overexpression in PCa remain to be elucidated.

Rizvi et al. recently reported that in cholangiocarcinoma, Yes-associated protein (YAP), a transcriptional coactivator and the key regulator of the Hippo pathway, interacts with TBX5, a transcription factor that binds to the promoter region of FGFR1 [15]. Hua et al. demonstrated that YAP regulates the expression of FGFR1 and acidic (FGF1) and basic (FGF2) FGF in ovarian cancer [16]. These observations provide possible mechanisms for FGFR1 upregulation; however, the potential involvement of these transcription factors in the development of resistance to cabozantinib, and in prostate cancer in general has not been previously studied.

The present study aims to examine the role of FGFR1 expression and activation in acquired resistance to MET/VEGFR2 inhibition by cabozantinib in PCa, demonstrate whether an FGF/FGFR1 autocrine loop serves as a compensatory mechanism for cell survival, and determine the role of YAP and TBX5 in the expression of FGFR1. We report that the transition from HGF/MET signaling to FGF/FGFR1 confers resistance to MET/VEGFR2 inhibition by cabozantinib. Our mechanistic studies demonstrated that the YAP/TBX5 complex is the key mediator for the overexpression of FGFR1 and the formation of an autocrine FGF/FGFR1 loop in PCa. Finally, the results from preclinical studies are confirmed in specimens from patients treated with cabozantinib.

## 2. Results

### 2.1. Increased FGFR1 Expression Mediates Acquired Resistance to MET Inhibition

We previously reported that prolonged inhibition of MET activity is associated with increased FGFR1 expression [8]. Here, we tested whether its most abundant ligands with the highest affinity, i.e., FGF1 and FGF2, were also upregulated in long-term cabozantinib-treated cells (42 days). Quantitative PCR analysis of cabozantinib-treated cells was performed. As shown in Figure 1, cabozantinib treatment significantly increased not only FGFR1 levels (Figure 1A), but also the mRNA levels of its ligands FGF1 and FGF2 (Figure 1B,C) in a dose-dependent manner.

To determine whether FGFR1 upregulation contributes to acquired resistance to cabozantinib, we first generated FGFR1-overexpressing (OV FGFR1) MDA PCa 144-13 cells. We previously showed that FGFR1 in MDA PCa 144-13 PDX was induced by cabozantinib [8]. FGFR1 expression was confirmed by Western blot (Figure 1D insert). FGFR1 overexpression had no effect on cell proliferation when compared with MDA PCa 144-13 cells transfected with a nontargeting (NT) vector, in vitro (Figure 1D). Inoculation of NT and OV FGFR1 cells into mice showed no difference in tumor growth (Figure 1E). We then examined the effect of cabozantinib treatment on the subcutaneous growth of these PDX tumors. For this experiment, mice were divided into four groups (NT, NT treated with cabozantinib, OV, OV treated with cabozantinib). Tumors were allowed to grow for 21 days to reach approximately 100 to 150 mm^3^ in size before initiation of treatment. While cabozantinib effectively inhibited tumor growth in NT xenografts, OV FGFR1 PDX grew exponentially in the presence of cabozantinib, at rates similar to the untreated tumors (Figure 1E). Cabozantinib-treated mice with tumors overexpressing FGFR1 had a considerably shorter survival than mice with NT tumors treated with cabozantinib (Figure 1F). Expression of FGFR1 in the OV FGFR1 tumors remained high at the end of the experiment, as determined by immunoblotting of tissue lysates (Figure 1G). As shown in Figure 1G, FGFR1 expression was further increased in cabozantinib-treated OV FGFR1 PDX, compared with untreated OV FGFR1 tumors [Figure 1G, short exposure (SE)]. We examined whether cabozantinib induces changes in vasculature in the tumors. As determined by IHC, cabozantinib treatment reduced CD31 expression in NT tumors but not in OV FGFR1 tumors (Figure 1H,I), suggesting that FGFR1 activation overcomes the antiangiogenic effect of MET/VEGFR2 inhibition. Taken together, these results suggest that FGFR1 overexpression is sufficient to confer resistance to cabozantinib treatment.

### 2.2. Cabozantinib Induces the Transcriptional Upregulation of YAP and TBX5

Next, we examined the molecular mechanism by which cabozantinib induces FGFR1 expression. The transcriptional coactivator YAP, together with the transcription factor TBX5, has been shown to regulate FGFR1 expression in other tumor types [15]. Thus, YAP and TBX5 are candidate transcription factors in the upregulation of FGFR1. We found that cabozantinib treatment increases YAP and TBX5 mRNA levels in a dose-dependent manner (Figure 2A,B). We then examined the effect of continuous cabozantinib treatment on the protein levels of YAP and TBX5. Immunoblotting was performed on lysates from MDA PCa 144-13 cells. As shown in Figure 2C,D, treatment with cabozantinib led to a time- and dose-dependent increase of YAP and TBX5 proteins relative to vehicle-treated controls. This increase correlates with a similar increase in the levels of FGFR1 and active FGFR1, pFGFR1 (Figure 2C,D).

To determine whether induction of YAP and TBX5 is due to MET inhibition by cabozantinib, we knocked down MET by infecting MDA PCa 144-13 cells with a sh-c-MET or a non-targeting (NT) virus. Knockdown of MET led to increased YAP and TBX5, as well as FGFR1 proteins, compared to the NT vector-transfected cells (Figure 2E). These observations support a potential role for YAP and TBX5 in the upregulation of the FGFR/FGF axis.

### 2.3. FGFR1, FGF1, and FGF2 Are Transcriptionally Upregulated Through a YAP/TBX5-Dependent Mechanism

The transcriptional coactivator YAP has been shown to form a complex with the transcription factor TBX5 to regulate the expression of several antiapoptotic genes [17]. The FGFR1 gene has been suggested as a target of YAP in cholangiocarcinoma, as chromatin immunoprecipitation assays demonstrated the simultaneous nuclear presence of TBX5-YAP complexes on the promoter region of FGFR1 [15]. Thus, we examined whether FGFR1 is upregulated through YAP and TBX5. PC3 cells displayed increased expression of FGFR1, YAP, and TBX5 compared to MDA PCa 144-13 cells (Figure 3A). As a result, we used PC3 cells to examine the roles of YAP and TBX5 in regulating FGFR1 expression. We used two different lentiviral-based shRNA constructs that specifically target YAP or TBX5 expression. PCR analysis showed that knockdown of YAP resulted in decreases in the mRNA levels of both YAP and TBX5 (Figure 3B,C). In contrast, knockdown of TBX5 resulted in a decrease in TBX5 mRNA but not in YAP mRNA (Figure 3B,C). Similar results were observed in protein levels (Figure 3D). These observations suggest that TBX5 expression is regulated by YAP. Importantly, decreased expression of YAP or TBX5 led to decreases in the mRNA levels of FGFR1, FGF1, and FGF2 (Figure 3E–G). Attenuation of YAP expression also resulted in decreased expression of other genes, e.g., GINS1 and RRM2 (Figure 3H), which are known to be downstream of YAP.

Another method of downregulating YAP is to use XAV939, a tankyrase inhibitor shown to suppress the transcription of YAP target genes [18]. We found that treatment of PC3 cells with XAV939 significantly decreased the expression of YAP and TBX5 in the nucleus as determined by immunofluorescence (Figure 4A), confirming the role of YAP in regulating TBX5 expression. Treatment of sh-c-MET MDA PCa 144-13 cells with XAV939 for 24 or 48 h also resulted in decreases in YAP and TBX5 protein in the nuclear fraction (Figure 4B) and FGFR1 protein in the cytoplasm fractions (Figure 4B).

To examine whether YAP and TBX5 form a complex to regulate FGFR1 expression, we used MDA PCa 144-13 cells in which YAP and TBX5 were upregulated by treating cells with cabozantinib (Figure 4C) or by transfection with shMET (Figure 4D). Immunoprecipitation of YAP in these cells identified the presence of TBX5 in YAP immunocomplexes (Figure 4D). Similarly, in PC3 cells, which expressed high levels of YAP, immunoprecipitation of YAP brought down TBX5 (Figure 4D). These results suggest a physical interaction between YAP and TBX5 in cells exhibiting high FGFR1 expression. Thus, it is likely that YAP and TBX5 form a complex to regulate FGFR1 expression, similar to those described in cholangiocarcinoma [15].

It is possible that increases in nuclear YAP protein levels are also due to a decrease in YAP phosphorylation, which stabilizes the protein and results in its nuclear localization. To examine such a possibility, we treated MDA PCa 144-13 cells with cabozantinib. We found that cabozantinib treatment did not decrease YAP phosphorylation at S127 or have any effect on the kinase activity of LATS1 or MST1/2 (Figure 4E), suggesting that YAP expression is most likely independent of the Hippo pathway. Together, these observations demonstrate that the mechanism of FGFR1 upregulation is an increase in nuclear YAP and TBX5 expression.

### 2.4. Downregulation of MET Signaling is Associated With Upregulation of FGFR1 Expression in AR-Positive PCa Cells

Having established the YAP/TBX5/FGFR1 axis in AR-negative PCa cell lines, we next determined whether cabozantinib treatment could elicit an increase in FGFR1 expression in AR-positive PCa cell lines. Hormone-sensitive (LNCaP) and enzalutamide-resistant (LREX) cell lines, which both express AR, were used (Figure 5A). Interestingly, LREX cells, which were generated from prolonged enzalutamide treatment of AR-overexpressing LNCaP cells (905 clone), showed upregulation of MET and glucocorticoid receptor (GR) compared to parental LNCaP cells (Figure 5A). 905L cells, which are LNCaP cells treated with vehicle during the LREX cell generation, were used as the control for LREX cells. Consistently, 905L cells showed similar MET, AR, and GR expression as LNCaP cells (Figure 5A). PC3 cells were used as a positive control. Western blot analysis of LREX cell lysates revealed that cabozantinib induced FGFR1 expression in a time-dependent manner (Figure 5B). Similarly, YAP and TBX5 proteins were also increased by cabozantinib treatment compared to vehicle-treated controls in LNCaP cells (Figure 5C). Knockdown of MET in LREX cells also led to increases in FGFR1, YAP, and TBX-5 protein expression compared to cells transfected with NT vector (Figure 5D). These findings indicate that cabozantinib treatment could also elicit an increase in FGFR1 expression in AR-positive PCa cell lines.

### 2.5. Cabozantinib Induces FGFR1 Activation and Increases in YAP and TBX5 Expression in PCa Bone Metastasis Specimens from a Clinical Trial

To investigate whether our findings might have relevance in human disease, we examined bone biopsy samples from patients with hormone-naïve metastatic PCa receiving ADT and cabozantinib. Trans-iliac bone marrow biopsies were obtained at baseline and after six weeks on cabozantinib. Specimens containing >5% tumor tissue from six patients were used in this study. Samples were stained for pFGFR, YAP, and TBX5. Expression of each protein was variable between patients at baseline, ranging from 0% to 75% for pFGFR, 0% to 25% for YAP, and 0% to 75% for TBX5. After six weeks on cabozantinib, there were relative increases in expression levels of pFGFR1, YAP, and TBX5 in 3/6, 4/6, and 4/6 patients, respectively. A representative example of IHC staining for each protein from one patient is shown in Figure 5E. While the limited number of patient samples precluded the association of expression status with clinical outcome, these data are in accordance with our preclinical findings and suggest that cabozantinib treatment drives a YAP/TBX5-mediated transcriptional increase in pFGFR1. The upregulation of these proteins after six-week cabozantinib treatment in some bone biopsies may suggest development of resistance. Collectively, the preclinical and human data support a model whereby cabozantinib treatment results in compensatory activation of FGF/FGFR1 signaling that in turn promotes resistance to cabozantinib.

## 3. Discussion

We report that upregulation of FGFR1, FGF1, and FGF2 confers acquired resistance to MET inhibition by cabozantinib. To the best of our knowledge, this is the first report to date that provides preclinical evidence demonstrating the transition in dependency from MET to FGFR1 signaling in PCa. Our in vivo experiments showed that FGFR1 overexpression is sufficient to bypass the growth-inhibitory effect of cabozantinib. Our data support the application of a combinatorial strategy, i.e., targeting both MET and FGFR1 signaling in order to derive a synergistic benefit and improve the clinical outcome in a subset of PCa patients. To this end, a novel oral MET/FGFR inhibitor has shown promising activity in patients with advanced solid tumors during a recent, proof-of-concept, phase I clinical trial [19].

Data regarding acquired resistance to MET inhibition are limited. Although activation of EGFR signaling [20], acquired MET mutations [21,22], and MET hyperamplification [23] have been reported as mechanisms of acquired resistance to MET TKIs, the link with FGFR1 signaling is still considered ambiguous. Two of the most common mechanisms of resistance to anticancer agents are drug efflux, which is mainly mediated by MDR-1 and inhibition of apoptosis/cell death, which is mainly executed by the upregulation of the antiapoptotic Bcl-2 family members. We examined the protein levels of MDR-1 and Bcl-2 in our model systems in response to cabozantinib and found that they are either not changed or even downregulated. Thus, these two mechanisms are unlikely to play a role in the acquisition of resistance to cabozantinib.

It has been reported that combined FGFR1 and MET inhibition can improve treatment efficacy in vitro and in xenografts of acute myeloid leukemia models [24]. In solid tumors, although activation of the FGFR1 pathway has been implicated in resistance to EGFR TKIs [25], a possible connection with MET TKIs has yet to be examined. A high-throughput secretome analysis has revealed that a shift from MET dependence to FGFR-2 and FGFR-3 dependence could lead to sustained cancer cell growth despite MET inhibition [26]. Using a different approach through synthetic lethal screening, Kim et al. showed that combined FGFR-3 and MET inhibition could surmount resistance to MET-targeting treatment [27]. Notably, our qPCR data suggest that FGFR1 transcriptional upregulation is associated with the emergence of acquired resistance to cabozantinib. This observation further supports the notion that, in the context of the low rate of genomic alterations in PCa, upregulation and hyperactivation of wild-type tyrosine kinases could drive tumor progression and enhance metastatic potential, underscoring the therapeutic potential of TKIs in the treatment of patients with PCa [28].

Our findings also shed light on a possible mechanism of FGFR1 overexpression in PCa. Although the role of FGFR1 signaling in PCa tumorigenesis and progression is well established [20,29], there are no data regarding the transcriptional mechanism that mediates FGFR1 expression. We demonstrated that, upon MET inhibition, FGFR1 upregulation is YAP/TBX5-dependent. Thus, we show that transcriptional coactivators (i.e., YAP), in conjunction with oncogenic transcription factors (i.e., TBX5), interact with molecular pathways and drive resistance to therapy. Recently, two studies identified a complex interaction between YAP and FGFR1 signaling. Rizvi et al. reported that in cholangiocarcinoma, YAP forms a complex with TBX5 to promote FGFR1 expression [15]. Similarly, Hua et al. demonstrated that overexpression of YAP drives upregulation of FGF ligands and FGFR1 in ovarian high-grade serous carcinoma [16].

Our study showed that YAP expression is increased upon MET inhibition. Previous studies have reported that there are Hippo-dependent (through YAP phosphorylation) and Hippo-independent mechanisms that regulate YAP activity [30]. We observed no change in the expression levels of p-YAP or the core kinases of the Hippo pathway. Our qPCR results suggested that upon MET inhibition, there is a transcriptional increase in YAP expression. However, the mechanism by which MET inhibition leads to an increase in YAP expression is unknown. There are only a few reports regarding regulation of YAP expression. Wu et al. reported that YAP is a target gene for GABP, a member of the Ets family of transcription factors [31]. Other studies identified, in the promoter region of the YAP gene, DNA sequences to which AP-1 and c-Jun transcription factors are able to bind [32]. In colon cancer cell lines, β-catenin regulates YAP mRNA expression, and chromatin-IP assays have revealed a β-catenin binding site in the promoter region of the YAP gene [33]. Interestingly, it has been reported that miR-375 targets YAP mRNA [34,35]. In particular, overexpression of miR-375 in PC3 cells, which display the lowest levels of miR-375 expression among PCa cell lines, has been shown to decrease YAP expression [36,37]. Taking into account the significance of both β-catenin [37,38] and mir-375 [38,39] in PCa, we postulate that these are candidate mechanisms that may account for the increase in YAP mRNA in our model system. Further studies are needed to delineate the mechanism that underlies the increase in YAP mRNA and protein expression in our PCa model system.

We found that TBX5 is involved in the expression of FGFR1 in cabozantinib-resistant clones. Although YAP has already been associated with resistance to treatment [37,40], our experiments imply that TBX5 is a critical mediator of resistance in PCa. To our knowledge, this is the first report that demonstrates a role for the transcription factor TBX5 in PCa. There is only one case report in the published literature linking TBX5 with the development of PCa, and the patient in the case report had Holt–Oram syndrome [41]. Moreover, our qPCR results suggest that YAP regulates TBX5 expression, although the mechanism is unknown. Our findings provide the first evidence that TBX5 is a mediator of FGFR1 expression in PCa.

The present study has identified YAP, TBX5, and activated FGFR1 as key regulators of resistance to MET-targeting treatment. In line with this observation, our IHC analysis demonstrated elevated expression of these proteins in human tissues from patients treated with the MET inhibitor cabozantinib compared with their expression in untreated tissue. The nuclear expression of both YAP and TBX5 in these clinical samples further supports a role for FGFR1 expression in cabozantinib resistance and suggests that nuclear TBX5 and YAP expression could be used as a biomarker to guide the application of FGFR-targeting treatment.

## 4. Materials and Methods

### 4.1. Cell Cultures

The human prostate cancer cell lines PC3 and LnCaP were maintained in Dulbecco’s modified Eagle medium with L-glutamine (DMEM/F12) and RPMI medium, respectively, supplemented with 10% fetal bovine serum (FBS) and 1% penicillin/streptomycin. MDA PCa 144-13 cells were maintained in suspension, as described previously [8,29,42]. PC3 and MDA PCa 144-13 cells are reported to be androgen receptor (AR)-negative [8,29], while LnCaP cells express AR. LREX and 905L cells were kindly provided by Charles Sawyers Lab and were maintained in RPMI with L-glutamine and HEPES buffer supplemented with 20% fetal bovine serum (Omega Scientific Inc, Tarzana, CA USA) and 1% penicillin/streptomycin. All cell lines were maintained at 37 °C and 5% CO_2_ and grown to 70% confluency. Routine testing for mycoplasma contamination was performed. Identity of cells was confirmed by fingerprinting analysis by the MD Anderson Cancer Center Department of Systems Biology, Core Facility.

### 4.2. Reagents

Cabozantinib (XL184) was provided by Exelixis (San Francisco, CA, USA). XAV939 was purchased from Sigma Aldrich (St. Louis, MO, USA).

### 4.3. Lentivirus-mediated YAP and TBX5 Silencing

To generate stable transfectants, PC3 or LnCaP cells were infected with lentivirus at the indicated multiplicity of infection (MOI) in the presence of polybrene, directing the expression of sh-YAP, sh-TBX5, or a non-targeting (NT) sequence (Appendix A). After 24 h, the lentivirus-containing medium was replaced by fresh medium, and the infected cells were selected with puromycin. NT, sh-YAP, and sh-TBX5 cells were sorted for red fluorescence protein. Western blotting and qRT-PCR assays were used to confirm the knockdown of YAP and TBX5 in these stable cell lines.

### 4.4. FGFR1 Overexpression

FGFR1 was overexpressed in MDA PCa 144-13 cells using the PCDNA3.1-FGFR1+P2A-eGFP expression plasmid (Appendix A). This plasmid was kindly provided by Nora Navone’s lab. The plasmid was transfected using the Lipofectamine 2000 reagent (Invitrogen, Carlsbad, CA, USA), according to manufacturer’s instructions. Transfected cells were selected for green fluorescence protein, and cell lysates were used for confirmation of FGFR1 overexpression by immunoblotting.

### 4.5. Proliferation Assays

Cell proliferation was determined by 3-(4,5-dimethylthiazol-2-yl)-2,5-diphenyltetrazolium bromide (MTT) assays (CellTiter 96^®^ Non-radioactive Cell Proliferation Assay, Promega, Madison, WI, USA). For the experiments, each cell type was seeded into a 96-well plate at a low density, in quintuplicate. Cells were incubated at 37 °C in an atmosphere of 5% CO_2._ All samples were processed according to the manufacturer’s instructions. Following the addition of the MTT solution, the cells were incubated for 3.5 h. Absorbance was then recorded at 570 nm using a 96-well plate reader (Envision 2104 multilabel reader, Perkin Elmer, Waltham, MA, USA). Three control wells with medium were used to obtain the blanks of absorbance.

### 4.6. Quantitative PCR Analysis

RNA was isolated from PC3 cells using RNAeasy™ mini kit (Qiagen, Germantown, MD, USA) and analyzed by quantitative RT-PCR, as previously described (Varkaris A G. S., 2013 Oct 1). *GAPDH* was used as a control. The primer sequences are listed in Appendix A.

### 4.7. Immunoblotting and Immunoprecipitation Assays

Cell lysates were prepared and immunoblotting was performed as described by Varkaris et al. [8]. Western blot analysis of tumor-bearing mouse tissue was performed as previously described [43]. Immunoprecipitation studies were performed using 1 mg of cell lysate, as described by Windham [44] et al. The antibodies used for these studies are listed in Appendix A.

### 4.8. Animal Studies

NOD/SCID mice were provided by MD Anderson Cancer Center, Experimental Radiation Oncology, Department of Veterinary Medicine & Surgery, authorized by the Jackson Laboratory, USA. All animal studies were conducted in accordance with the current regulations and standards of the U.S. Department of Agriculture, the U.S. Department of Health and Human Services and the National Institutes of Health and were approved by The University of Texas MD Anderson Cancer Center Institutional Animal Care and Use Committee (00000840-RN02).

For the cabozantinib resistance study, 100 µL of PBS containing 1x10^6^ MDA PCa 144-13 cells were subcutaneously injected into the right flank of 5- to 6-week-old mice. Tumors were allowed to grow until they reached a volume of 100 to 150 mm^3^. Then tumor-bearing mice were randomized so that each group had an equal number of animals with similar tumor size. Mice were treated daily by oral gavage with sterile H_2_O (5 µL/g for withdrawal, control) or cabozantinib (30 mg/kg body weight in 5 µL/g sterile H_2_O).

All animals were regularly monitored for changes in their health status and weight. Tumor size was measured with calipers three times/week and tumor volume was calculated with the formula: V = ½ (*length* × *width*^2^) [45]. Mice were euthanized when tumor volume exceeded 1500 mm^3^, or if they had weight loss exceeding 30%. Tumor samples were harvested for immunoblotting, and immunohistochemistry, and immunofluorescence staining as previously described [8].

### 4.9. Immunohistochemistry

Immunohistochemical analysis of human samples was performed as described previously [8]. Briefly, 4-µm paraffin sections were cut and placed in the oven for one hour at 60 °C. The sections were cooled to room temperature for five minutes prior to deparaffinization and rehydration (xylene 2×, 100% EtOH 2×, 95% EtOH 2×, 80%EtOH 1×, PBS 3×). Tissue sections were pretreated in citrate buffer solution, pH 6.1 (Dako, Carpinteria, CA, USA), and heated in a Pascal pressure cooker (Dako) (set at 125 °C for 30 s) to unmask the epitopes. The sections were rinsed three times in PBS followed by a rinse in a PBS/0.05% Tween 20 solution (pH 8.0). Endogenous peroxidases were blocked with Dual Endogenous Enzyme Block (Dako) for ten minutes, followed by three rinses in PBS followed by one rinse in a PBS/0.05% Tween 20 solution (pH 8.0). The sections were incubated with serum-free protein block for one hour at room temperature, followed by incubation with the primary antibodies, listed in Appendix A, overnight at 4 °C. The sections were rinsed three times in PBS, followed by a rinse in a PBS/0.05% Tween 20 solution (pH 8.0), and incubated with the Dako EnVision+ Dual Link System-HRP for one hour. This step was followed by incubation with the DAB substrate (Dako). The sections were counterstained for one minute with Mayer’s hematoxylin (Poly Scientific, Bayshore, NY, USA) diluted 1:10 with deionized H_2_O, followed by two H_2_O rinses. The nuclei were stained with bluing reagent (Richard Allen Scientific, Sugar Land, TX, USA) for one minute and were then dehydrated through xylene and increasing percentages of EtOH (95% EtOH 3×, 100% EtOH 3×, xylene 3×). The sections were then covered with a glass coverslip using a xylene-based mounting media and air dried at room temperature.

To quantify angiogenesis and examine resistance to cabozantinib from a different perspective, microvessel density (MVD) was determined by vascular structures that stained positive for CD31. Fifteen images per tumor were captured, and CD31-positive vessels were manually counted.

### 4.10. Human Tissues

Trans-iliac bone marrow biopsies containing tumor tissue were obtained at baseline and after six weeks on cabozantinib in men with hormone-sensitive metastatic prostate cancer receiving androgen deprivation therapy (ADT) plus cabozantinib on a phase II clinical trial conducted at our institution (clinicaltrials.gov ID NCT01630590). A full report of this study will be published in a separate manuscript (Corn PG, Zhang M, Phase 2 Study of Cabozantinib and Androgen Ablation in Patients with Hormone-Naïve Metastatic Prostate Cancer, Clin Can Res, Accepted 2019 October 17, in press.) but is briefly summarized here. In this study, 62 patients received cabozantinib (starting dose 60 mg PO daily) and ADT with a GnRH analogue. Patients could receive up to three months of ADT prior to initiating cabozantinib. Bone scan (BS) and CT scans were performed every twelve weeks. Efficacy endpoints included response in markers, BS and RECIST, and time to castrate-resistant progression defined by radiographic progression, clinical progression, or receipt of additional anti-cancer therapy. Trans-iliac bone marrow biopsies obtained at baseline and after six weeks on cabozantinib treatment containing >5% tumor were available for six patients.

### 4.11. Statistics

Differences among groups were assessed by analysis of variance (ANOVA) followed by Fisher’s exact test for multiple comparisons. Tumor growth was compared by repeated-measures ANOVA and survival by the log-rank (Mantel–Cox) test. *P* values < 0.05 were considered significant. Error bars show means ± standard error of the mean (SEM). For the study of the effect of cabozantinib treatment on the subcutaneous growth of these PDX tumors, seven mice were used per group. To study the effect of sh-YAP and sh-TBX5 on tumor growth in vivo, six mice were used per group.

## 5. Conclusions

In conclusion, our data, based on the integration of preclinical and clinical experiments in cell lines, PDXs, and human tissue, reveal a novel link between MET inhibition and a YAP/TBX5-mediated increase in FGFR1 expression (Figure 5F). Our findings will trigger a reevaluation of the appropriate clinical strategy for MET-targeted therapy. Our data link the experimentally defined mechanism of FGFR1 expression in murine systems to clinical observations. These parallel investigations provide insight into the mechanism and identify multiple potential targets that may lead to biologically informed clinical trials. Improving the understanding of the mechanism of RTK expression will provide predictive biomarkers that will inform the clinical application of TKIs.

## Figures and Tables

**Figure 1 cancers-12-00244-f001:**
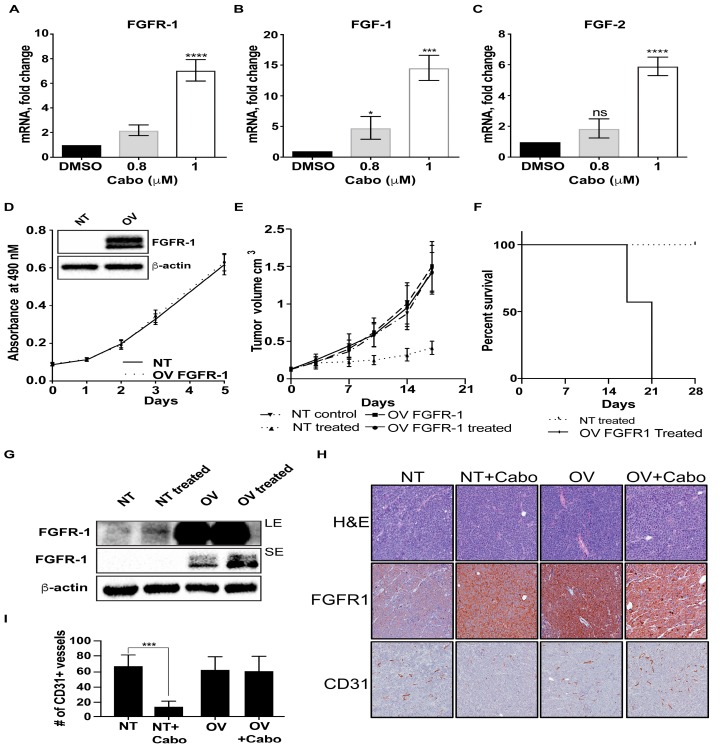
FGFR1 overexpression mediates acquired resistance to MET/VEGFR2 inhibition. (**A**–**C**) Cabozantinib (Cabo) induces the transcriptional upregulation of FGFR1, FGF1, and FGF2. MDA PCa 144-13 cells were treated continuously with the indicated doses of cabozantinib in vitro. Relative FGFR1, FGF1, and FGF2 mRNA expression was analyzed by qRT-PCR. GAPDH was used as a control. Results are expressed as fold change compared to vehicle-treated cells. Columns represent mean values ± SEM. (**D**) Overexpression of FGFR1 in MDA PCa 144-13 cells. Western blotting for FGFR1 expression. Whole-cell extracts obtained from MDA PCa 144-13 cells transfected with a nontargeting (NT) vector or an overexpressing (OV) FGFR1 vector were analyzed. (**E**) Growth of NT and OV FGFR1 cells in vivo. Mice were treated by oral gavage with 30 mg/kg cabozantinib or H_2_O for indicated times. (**F**) Modified Kaplan-Meier curve of mice with OV FGFR1 and NT tumors treated with cabozantinib. Statistical analysis by log-rank (Mantel–Cox) test for time to 1500 mm^3^ tumor volume. (**G**) Immunoblot analysis for FGFR1 expression of MDA PCa 144-13 tumor lysates from NT, NT cabozantinib-treated, OV, and OV cabozantinib-treated tumors. LE, long exposure. SE, short exposure. **(H**,**I)** Effect of cabozantinib treatment on vasculature. H&E staining and immunohistochemistry of FGFR1 and CD31 expression in NT and OV MDA PCa 144-13 tumors in the presence and absence of cabozantinib are compared. * *p* < 0.05; *** *p* < 0.01; **** *p* < 0.001. More details of western blot, please view at the Appendix A.

**Figure 2 cancers-12-00244-f002:**
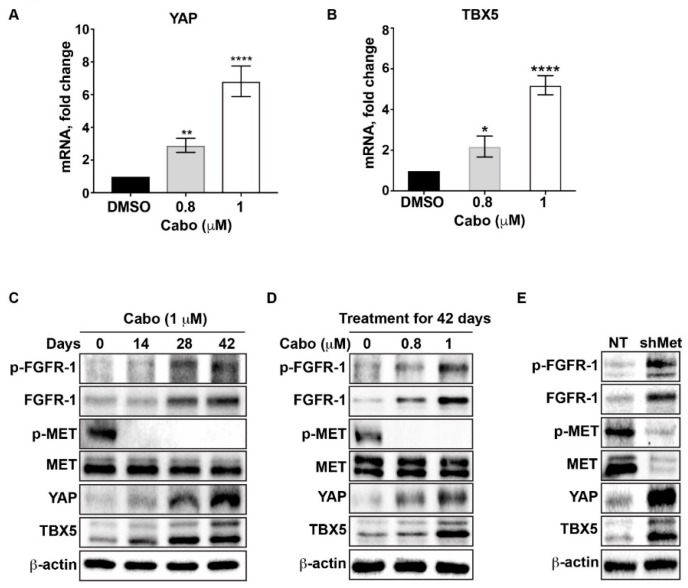
Cabozantinib induces the upregulation of YAP and TBX5. (**A**,**B**) MDA PCa 144-13 cells were treated continuously with the indicated doses of cabozantinib in vitro. YAP and TBX5 mRNA expression were analyzed by qRT-PCR. GAPDH was used as a control. Results are expressed as fold change compared to vehicle-treated cells. Columns represent mean values ± SEM (* *p* < 0.05; ** *p* < 0.01; **** *p* < 0.001). (**C**) Acquired resistance to cabozantinib is associated with increased phosphorylation of FGFR1 and increased FGFR1, YAP, and TBX5 protein expression. MDA PCa 144-13 cells were treated with cabozantinib in vitro and the effect on MET and FGFR1 activity and YAP and TBX5 expression was examined. Immunoblot analysis of cell lysates was performed on control cells and at 14, 28, and 42 days of treatment. (**D**) MDA PCa 144-13 cells were treated with the indicated concentrations of cabozantinib for 42 days. (**E**) Immunoblot of NT-transfected and sh-met-transfected MDA PCa 144-13 cells. More details of western blot, please view at the Appendix A.

**Figure 3 cancers-12-00244-f003:**
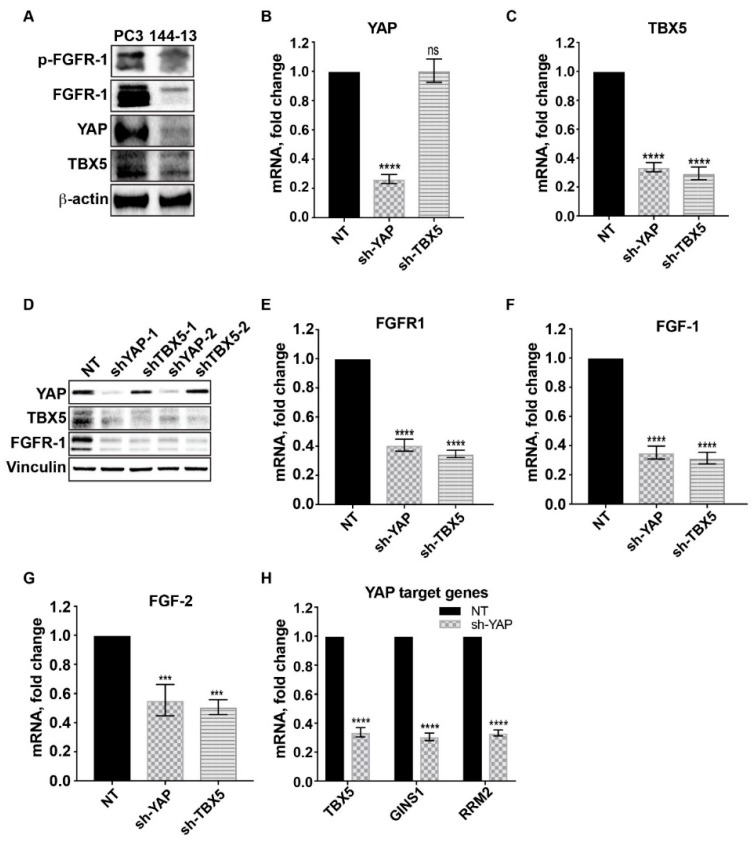
FGF and FGFR1 expression is induced by a YAP/TBX5-dependent mechanism. (**A**) Immunoblot of MDA PCa 144-13 and PC3 cells showed that PC3 expressed high levels of FGFR1 accompanied with high levels of YAP and TBX5 compared to MDA PCa 144-13 cells. (**B**,**C**) qRT-PCR for relative YAP and TBX5 mRNA levels in PC3 cells transfected with non-targeting (NT) or shRNA constructs targeting YAP and TBX5. Sh-YAP and sh-TBX5 knocked down YAP and TBX5 effectively. Knockdown of YAP also reduced TBX5 mRNA. However, knockdown of TBX5 did not affect YAP mRNA. (**D**) FGFR1 expression was evaluated by immunoblotting upon YAP and TBX5 silencing with two different shRNA constructs (1 and 2) in PC3 cells. (**E**–**G**) qRT-PCR for relative FGFR1, FGF1, and FGF2 mRNA expression in PC3 cells transfected with non-targeting (NT) or shRNA constructs targeting YAP and TBX5. (**H**) qRT-PCR for TBX5 and known YAP target genes (GINS1 and RRM2) mRNA expression in NT-transfected and sh-YAP–transfected PC3 cells. Results are expressed as fold change compared to NT-transfected cells. Columns represent mean values ± SEM (*** *p* < 0.001; **** *p* < 0.0001). More details of western blot, please view at the Appendix A.

**Figure 4 cancers-12-00244-f004:**
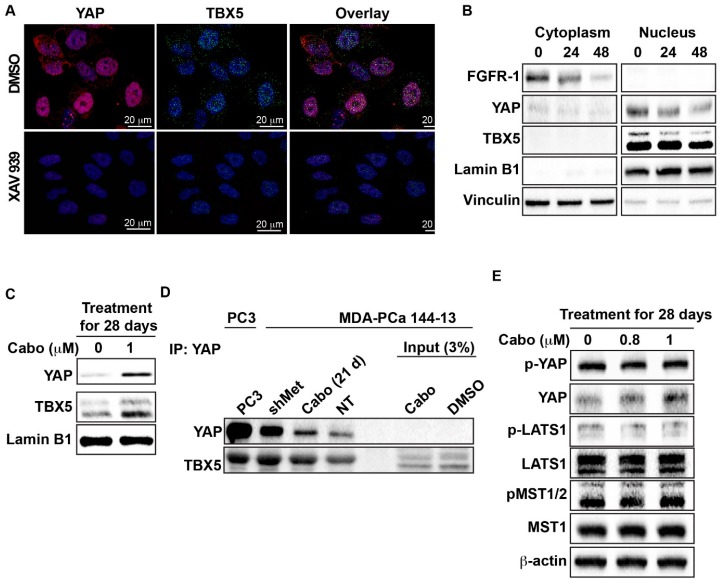
Nuclear localization of YAP and TBX5 and their complex formation. (**A**) Representative confocal microscopy images showing immunofluorescence of YAP (red, Alexa647), TBX5 (green, Alexa488), and DAPI (blue) in control and XAV939-treated PC3 cells; (**B**) sh-c-MET MDA PCa 144-13 cells were treated with XAV939 for 24 or 48 h. Nuclear and cytoplasmic fractionation was performed and lysates were immunoblotted for FGFR1, YAP, and TBX5 expression. (**C**) MDA PCa 144-13 cells were treated with or without cabozantinib. Nuclear extracts were prepared, and YAP and TBX5 expression were determined by immunoblotting; (**D**) Interaction between YAP and TBX5. Cell lysates from PC3 and MDA PCa 144-13 (NT-transfected, sh-met–transfected, cabozantinib-treated) were prepared and immunoprecipitated with YAP antibody or control IgG. Immunoblots of input lysates (3% of inputs) of DMSO and cabozantinib-treated MDA PCa 144-13 cells are also shown. (**E**) MDA PCa 144-13 cells were treated for 28 days with the indicated cabozantinib concentrations and the effect on key components of the Hippo signaling pathway was examined. Immunoblot analysis of whole cell lysates was performed, and p-YAP (S127), YAP, pLATS1, LATS, pMST1/2, and MST1 expressions were determined. More details of western blot, please view at the Appendix A.

**Figure 5 cancers-12-00244-f005:**
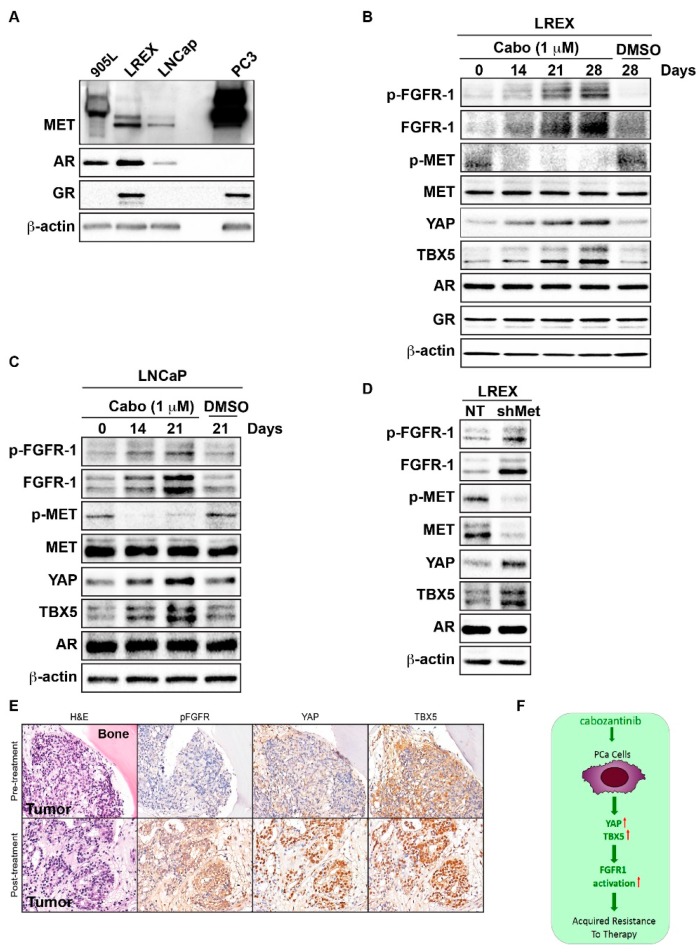
MET inhibition is associated with increased phosphorylation of FGFR1 and increased Figure. YAP, and TBX5 protein expression in AR-positive PCa models. (**A**) Whole-cell extracts obtained from 905L (LREX parental cell line), LREX, LNCaP, and PC3 cells were analyzed by immunoblotting for MET, AR, and GR protein expression. (**B**) LREX cells were treated with cabozantinib, and the effect on MET and FGFR1 activity and YAP, TBX5, AR, and GR expression was examined. Immunoblot analysis of cell lysates was performed on control cells and at 14, 21, and 28 days of treatment at indicated cabozantinib concentration. (**C**) LNCaP cells were treated with cabozantinib and the effect on MET and FGFR1 activity and YAP, TBX5, and AR expression was examined. Immunoblot analysis of cell lysates was performed on control cells and at 14, 21, and 28 days of treatment at indicated cabozantinib concentration. (**D**) Immunoblot of LREX, NT-transfected, and sh-met–transfected cells. (**E**) Effect of cabozantinib on pFGFR, YAP, and TBX5 expression in human PCa tumors. H&E staining and immunohistochemical evaluation of pFGFR, YAP and TBX5 expression in matched samples (trans-iliac bone marrow biopsies) from a patient with prostate cancer, obtained at baseline and after six weeks under cabozantinib treatment. (**F**) Schematic representation summarizing our findings on the molecular mechanism of resistance to cabozantinib. More details of western blot, please view at the Appendix A.

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
