# Peer review of "Resistance to MET/VEGFR2 Inhibition by Cabozantinib Is Mediated by YAP/TBX5-Dependent Induction of FGFR1 in Castration-Resistant Prostate Cancer"

_cancers, 2020, doi:10.3390/cancers12010244_

Round 1

Reviewer 1 Report

The paper by Koinis F and coworkers the molecular mechanisms responsible for prostate cancer resistance to cabozantinib is evaluated. The authors demonstrate that resistance to MET/VEGFR2 inbibition by the drug is associated to YAP/TBX5 dependent overexpression of FGFR1 autocrine pathway. 

The experiments are appropriately designed and results sound. 

I have only minor comments to improve the manuscript.

1) the authors report the mouse experiments with xenograft of prostate cells as PDX. I suggest to use the general term "xenograft" since the cells used are cell lines heterotopically implantanted by subcutaneous injection. moreover the sorurce of MDA PCa 144-13 cells is not reported in the MM section.

2) the sentence at line 100-102 shlould be rewritten substituting "when mice were sacrified" with "at the end of treatment" or "at the end of the experiment".

3) The pictures in the composition of figure 1H are not at high definition. Please enlarge this panel in order to appreciate the quality of the images.

Author Response

The paper by Koinis F and coworkers the molecular mechanisms responsible for prostate cancer resistance to cabozantinib is evaluated. The authors demonstrate that resistance to MET/VEGFR2 inbibition by the drug is associated to YAP/TBX5 dependent overexpression of FGFR1 autocrine pathway. 

The experiments are appropriately designed and results sound. 

Our response: We thank the reviewer for the accurate summary of our manuscript and the positive comments

I have only minor comments to improve the manuscript.

1) the authors report the mouse experiments with xenograft of prostate cells as PDX. I suggest to use the general term "xenograft" since the cells used are cell lines heterotopically implantanted by subcutaneous injection. moreover, the source of MDA PCa 144-13 cells is not reported in the MM section.

Our response: The MDA PCa 144-13 is a patient derived xenograft (PDX) that has been developed by the PDX core center at the MD Anderson Cancer center and has been previously described (Aparicio A et al., Prostate, 2011, Tzelepi et al., 2012, Clinical Cancer Research and in our most recent publication, Varkaris et al., 2016, Clinical Cancer Research). We have now introduced the additional references in the manuscript (line 367).

2) the sentence at line 100-102 should be rewritten substituting "when mice were sacrified" with "at the end of treatment" or "at the end of the experiment".

Our response:  This has now been corrected according to the reviewer’s suggestion (line 110)

3) The pictures in the composition of figure 1H are not at high definition. Please enlarge this panel in order to appreciate the quality of the images.

Our response:  We have now introduced the high-resolution images in a bigger panel as part of Figure 1H and the quantification as figure 1I (Line 115 and 132).

Reviewer 2 Report

Authors suggest that YAP/TBX5-dependent induction of FGFR1 mediates resistance to MET and VEGFR2 inhibition by cabozantinib in prostate cancer. Despite interesting data, it has some concerns as follows:

In general it is well organized and written. How do you think FGFR1 is a major target for cabozantinib. resistance directly or indirectly? How about effect on BCl2 and MDR1? Above all, show the features of prostate cancer cells such as PC3 and MDA PCa 144-13 cells. Are MDA PCa 144-13 cells castration nonresistant prostate cancer cell lines? How about cytotoxicity of cabozantinib in PC3, LNCap and MDA PCa 144-13 cells, DU145 cells? How about effect of FGFR1 overexpression or knockdown on MDR1 and BCL2 in PC3 and MDA PCa 144-13 cells? Show the cabozantinib. Resistance such as MDR1 and BCL2 in PC3 or castration dependent cells compared to nonresistant prostate cancer cells But define several abbreviations including MET, VEGFR2, FGFRs, PDGFR, HER2 How about binding between Yap or and FGFR1 in PC3 cells

   Minor

Check grammar; YAP and TBX5 mRNA 144 expression were analyzed by qRT-PCR.

Author Response

Please see attached file for details response to the questions/comments. 

Reviewer 3 Report

interesting paper entitled “Resistance to MET/VEGFR2 Inhibition by 2 Cabozantinib is Mediated by YAP/TBX5-Dependent 3 Induction of FGFR1 in Castration-resistant Prostate 4 Cancer”.

I have some suggestions for the authors, that I hope could contribute to improve this paper before the publication in Cancers Journal. 

Minor issues

- According to The MIQE Guidelines (Bustin et al., 2009), more than one housekeeping gene should be analysed for their expression stability in RT-qPCR experiments, in order to choice the more stable reference gene suitable for normalization.

Bustin, S. A. et al. The MIQE guidelines: minimum information for publication of quantitative real-time PCR experiments. Clin Chem. 55, 611–622 (2009).

-Concerning immunoblotting experiments, the authors should explain the presence of two protein bands (MET, TBX5, p-FGFR1) shown in some samples.

- The authors should state the significance of “cabo” abbreviation reported in Figure 1.

- The term ”b-catenin” should be replaced with “beta-catenin” or “b-catenin”

Author Response

Please see attached file for the detailed response to the reviewer's questions/comments

Round 2

Reviewer 2 Report

They addressed well to my comments